# Pharmacological Potential and Chemical Composition of *Crocus sativus* Leaf Extracts

**DOI:** 10.3390/molecules27010010

**Published:** 2021-12-21

**Authors:** Olha Mykhailenko, Vilma Petrikaite, Michal Korinek, Fang-Rong Chang, Mohamed El-Shazly, Chia-Hung Yen, Ivan Bezruk, Bing-Hung Chen, Chung-Fan Hsieh, Dmytro Lytkin, Liudas Ivanauskas, Victoriya Georgiyants, Tsong-Long Hwang

**Affiliations:** 1Department of Pharmaceutical Chemistry, National University of Pharmacy of Ministry of Health of Ukraine, 4-Valentinivska St., 61168 Kharkiv, Ukraine; mykhailenko.farm@gmail.com (O.M.); vania.bezruk@gmail.com (I.B.); vgeor@nuph.edu.ua (V.G.); 2Laboratory of Drug Targets Histopathology, Institute of Cardiology, Lithuanian University of Health Sciences, Sukilėlių pr. 13, LT-50162 Kaunas, Lithuania; vilma.petrikaite@lsmuni.lt; 3Life Sciences Center, Institute of Biotechnology, Vilnius University, Saulėtekio al. 7, LT-10257 Vilnius, Lithuania; 4Graduate Institute of Natural Products, College of Pharmacy, Kaohsiung Medical University, Kaohsiung 80708, Taiwan; michalk@kmu.edu.tw (M.K.); aaronfrc@kmu.edu.tw (F.-R.C.); chyen@kmu.edu.tw (C.-H.Y.); 5Department of Biotechnology, College of Life Science, Kaohsiung Medical University, Kaohsiung 80708, Taiwan; bhchen@kmu.edu.tw; 6Graduate Institute of Natural Products, College of Medicine, Chang Gung University, Taoyuan 33302, Taiwan; 7Research Center for Chinese Herbal Medicine, Research Center for Food and Cosmetic Safety, Graduate Institute of Health Industry Technology, College of Human Ecology, Chang Gung University of Science and Technology, Taoyuan 33302, Taiwan; 8Department of Pharmacognosy, Faculty of Pharmacy, Ain-Shams University, Organization of African Unity Street, Abassia, Cairo 11566, Egypt; 9Department of Pharmaceutical Biology, Faculty of Pharmacy and Biotechnology, German University in Cairo, Cairo 11835, Egypt; 10The Institute of Biomedical Sciences, National Sun Yat-sen University, Kaohsiung 80424, Taiwan; 11Department of Biochemistry and Molecular Biology, College of Medicine, Chang Gung University, Taoyuan 33302, Taiwan; doctordoctorkk@yahoo.com.tw; 12Educational and Scientific Institute of Applied Pharmacy, National University of Pharmacy of Ministry of Health of Ukraine, 12-Kylikivska St., 61000 Kharkiv, Ukraine; d.v.lytkin@gmail.com; 13Department of Analytical and Toxicological Chemistry, Lithuanian University of Health Sciences, A. Mickevičiaus g. 9, LT-44307 Kaunas, Lithuania; liudas.ivanauskas@lsmuni.lt; 14Department of Anesthesiology, Chang Gung Memorial Hospital, Taoyuan 33305, Taiwan; 15Department of Chemical Engineering, Ming Chi University of Technology, New Taipei City 24301, Taiwan

**Keywords:** *Crocus sativus*, cytotoxic activity, free radical scavenging activity, HPLC fingerprint, anti-allergic activity, anti-inflammatory activity, anti-viral activity

## Abstract

*Crocus sativus* L. (saffron) has been traditionally used as a food coloring or flavoring agent, but recent research has shown its potent pharmacological activity to tackle several health-related conditions. *Crocus* sp. leaves, and petals are the by-products of saffron production and are not usually used in the medicine or food industries. The present study was designed to determine the chemical composition of the water and ethanolic extracts of *C. sativus* leaves and test their cytotoxic activity against melanoma (IGR39) and triple-negative breast cancer (MDA-MB-231) cell lines by MTT assay. We also determined their anti-allergic, anti-inflammatory, and anti-viral activities. HPLC fingerprint analysis showed the presence of 16 compounds, including hydroxycinnamic acids, xanthones, flavonoids, and isoflavonoids, which could contribute to the extracts’ biological activities. For the first time, compounds such as tectoridin, iristectorigenin B, nigricin, and irigenin were identified in *Crocus* leaf extracts. The results showed that mangiferin (up to 2 mg/g dry weight) and isoorientin (8.5 mg/g dry weight) were the major active ingredients in the leaf extracts. The ethanolic extract reduced the viability of IGR39 and MDA-MB-231 cancer cells with EC_50_ = 410 ± 100 and 330 ± 40 µg/mL, respectively. It was more active than the aqueous extract. Kaempferol and quercetin were identified as the most active compounds. Our results showed that *Crocus* leaves contain secondary metabolites with potent cytotoxic and antioxidant activities.

## 1. Introduction

Cancer, one of the most devastating and complex diseases, is tackled by several techniques, including radiotherapy, immunotherapy, and chemotherapy. However, the toxicity and resistance of synthetic chemotherapeutic drugs have prompted scientists to search for natural chemotherapeutic agents obtained from plants and marine organisms. Since the 1970s, great progress has been made in the search for and development of natural anticancer drugs derived from medicinal herbs [1]. About 35,000 plant species were tested by the National Cancer Institute (NCI) for their potential anti-tumor activity, and among them, about 3000 plant species showed reproducible anti-tumor activity [2]. In addition to their potent anticancer effect, natural chemotherapeutics agents serve as drug leads for the vast majority of synthetic chemotherapeutic agents [2]. Anticancer drugs of plant origin proved to exhibit a safer safety profile in comparison with their synthetic counterparts but were less selective [3]. The interesting therapeutic properties of natural chemotherapeutic agents have encouraged scientists to search for more natural cytotoxic drugs from medicinal or edible plants. Among the edible plants that attracted attention in the last few decades is *Crocus sativus* (saffron).

*Crocus sativus* L. is a perennial underground corm that belongs to the family *Iridaceae*. In Ayurveda, saffron is used to treat cold, cough, asthma, arthritis, acne, and skin diseases and as an aphrodisiac and stimulant. In ancient Egypt, saffron was used to treat all types of gastrointestinal diseases. Saffron was an ingredient of *laudanum* or *tinctura opii crocata*, a “Swedish herbal mixture” for digestion [4]. Saffron is used in folk medicine to treat cancer, asthma, hysteria, convulsion, and spasms [5]. In Ukraine, saffron is used as an anticancer agent and for the treatment of eye diseases [6]. Saffron is used in oriental medicine in central and south Asia [7], and it is included in the European Pharmacopoeia as a homeopathic remedy [8].

*Crocus* stigmas contain many important secondary metabolites, such as crocin, crocetin, safranal, and picrocrocin, which showed potent cytotoxic and antiproliferative activity in different models and many cancer cell lines [9,10,11]. Recent scientific research indicated the potential application of *Crocus* stigma supplements for retinal treatment [12] and to improve metabolism in obese people [13]. Several pharmacological activities were reported for saffron and its constituents, including antioxidant, antinociceptive, anti-inflammatory, anticonvulsant, antidepressant, antitumorigenic, and immunomodulatory effects [14].

In addition to *Crocus* stigmas, the saffron plant has many by-products, including leaves, petals, and corms. To collect 1 kg of stigmas, about 350 kg of petals, 1500 kg of leaves, and the corms are not used and thrown away [15]. However, this biomass can be used as a potentially significant source of biologically active secondary metabolites, including flavonoids and anthocyanins. The exploitation of these by-products can significantly increase the profitability and sustainability of saffron production [16,17]. However, the data on the biological activity, especially the cytotoxic effect, of other *C. sativus* organs are still scarce.

In one report, it was shown that the proliferation of human adenocarcinoma cells (Caco-2) was greatly inhibited by the ethanolic extracts of *C. sativus* petals and leaves (EC_50_ = 0.42 mg/mL), while the corm extract caused some signs of toxicity and completely abolished proliferation (EC_50_ = 0.05 mg/mL) [15].

Mir et al. [18] reported the isolation of naturally derived crocetin *β*-D-glucosyl ester from the leaves of *C. sativus* var. *cashmerianus* obtained from Pampore Kashmir, India. The purified compound showed an antiproliferative effect against the human breast adenocarcinoma cell model (MCF-7), possibly by inhibiting the estrogen receptor alpha and HDAC2-mediated signaling cascade. The extracts of *Crocus* leaves showed antibacterial, antioxidant [19], and metal ion chelating activities [20] and mild antidepressant and anti-inflammatory efficacy [21]. The main aim of the current study was to investigate the phytochemical content of the water and ethanolic extracts from *Crocus* leaves grown in Ukraine and to determine their cytotoxic activity on several cancer cell lines, including melanoma (IGR39) and triple-negative breast cancer (MDA-MB-231) cells. Until now, the extracts of *C. sativus* leaves grown in Ukraine have not been previously studied for their cytotoxic potential.

## 2. Results and Discussion

*Crocus* waste products, such as the leaves and flowers, are gaining popularity as potential cytotoxic drugs. However, there is a lack of empirical data to support this claim. In this study, the activities of the aqueous and ethanolic extracts of *Crocus* leaves from Ukraine were evaluated for their cytotoxic activity and other bioactivities, including antiviral, antiallergic, and anti-inflammatory effects. The composition of the biologically active substances in the plant extracts may depend on the selected extraction method, and, most importantly, on the extractant [22], and thus we prepared the ethanolic and aqueous extracts for the selected type of raw material to be able to compare their chemical compositions. The solvent extraction method was chosen according to the most commonly used method [23]. In this study, the crude extracts of *C. sativus* leaves were prepared using water-ethanol (70%) and distilled water as the solvents. The effect of each extract on the viability of melanoma (IGR39) and triple-negative breast cancer (MDA-MB-231) cells was determined using the MTT assay.

### 2.1. Qualitative and Quantitative Analysis of the Identified Compounds

Previous phytochemical analysis of *C. sativus* leaves from Ukraine showed the presence of flavonoids (*trans*-cinnamic acid, chlorogenic acid, kaempferol, ononin, irigenin, and mangiferin), as well as amino acids and carboxylic acids in the plant raw material [24]. Therefore, further research was devoted to establishing the chemical composition of the dry *C. sativus* leaf extracts aiming to reveal their effect on the pharmacological activity of the extracts.

To determine the composition of the tested extracts, HPLC was used. The HPLC fingerprints of *Crocus* aqueous and ethanolic dry extracts are shown in Figure 1 and Table 1. All peaks in the chromatograms were identified by comparing their chromatograms with the chromatograms of the reference compounds. The compound structures were further elucidated based on their fragmentation pathways. UPLC–MS/MS analysis was applied to all the phenolic standards to identify and confirm each compound’s chemical structure. Since polyphenols contain one or more hydroxyl and/or carboxylic acid groups, MS data were acquired in the negative ionization mode. The identification of the phenolic compounds was carried out by comparing retention times (tR), calculated molecular weights, and MS/MS data of the 16 reference standards. The full spectroscopic data of the compounds are shown in Table 2. A brief description and a comparison with the data reported compounds in previous literature are provided in the Appendix A.

*Phenol carboxylic acid derivatives.* According to the literature data, some phenol carboxylic acids (caffeic, chlorogenic, and gallic acid) were only found in the stigma aqueous extract of *C. sativus* [25]. Acar et al. [26] identified *p*-coumaric acid and rosmarinic acid in *C. baytopiorum* leaves. No more literature data were found regarding the study of phenol carboxylic acids in the leaves of *Crocus* sp. Thus, in this study, the presence of chlorogenic acid and caffeic, ferulic, and cinnamic acids in *C. sativus* leaf extracts was established for the first time (Table 1). The content of the chlorogenic acid in the aqueous and ethanolic extracts of *Crocus* leaves was the same (0.68 mg/g). Cinnamic acid, a precursor in the biosynthesis of flavonoids via the shikimate pathway, was only found in the ethanolic extract of *Crocus* leaves (1.42 mg/g) [27,28]. Our data indicated that *Crocus* leaves contained more derivatives of hydroxycinnamic acids and fewer flavonoids.

*Xanthones*. It is known that plants of the genus *Iris* (*Iridaceae*) accumulate various xanthones, mainly C-glycosylxanthones [29]. In the genus *Crocus*, mangiferin was isolated and only identified from the leaves of *C. aureus* and *C. stellaris* [30,31,32]. Therefore, mangiferin can be considered one of the markers of this family. On the chromatograms, peak C at 17.18 min was identified as mangiferin based on comparison with the UV spectrum of the reference compound. This is the first report on the identification of mangiferin from *C. sativus* leaf extracts. In *Crocus* leaves, the content of mangiferin was almost equal in both extracts (1.8–2.0 mg/g).

*Flavonoids and isoflavonoids and their glycosides*. Previous reports showed the presence of glycosylated derivatives of luteolin, kaempferol, quercetin, and apigenin in *C. sativus* leaf ethanolic extract [33]. Flavonoid glycosides are better recovered by water [34,35]. Isoorienin (luteolin-6-C-glucoside) and genistein-7-O-glucoside were identified in this experiment in the aqueous extract only of *Crocus* leaves. In the leaves, the content of the isoflavone glycoside tectoridin (7-glucoside tectorigenin) was higher in the ethanolic extract (0.11 mg/g).

It should be noted that the flavonoid aglycones are more hydrophobic [36] than their corresponding glycosides, and thus their solubility is low. The solubility of quercetin and kaempferol in water is very limited. Therefore, the concentration of kaempferol in the aqueous extract was low (0.033 mg/g) in comparison with the ethanolic extract that was slightly higher (0.046 mg/g).

Tectoridin and iristectorigenin B were previously only isolated or identified from the genus *Iris* [29,32]. This is the first report on the identification of those compounds from *C. sativus* leaves. Among the detected isoflavonoid anglycones, iristectorigenin B, nigricin, irigenin, and biochanin A were identified. These compounds possess from 1 to 3 hydroxy groups and 0–3 methoxy groups, which affects their solubility in the extracting solvent. The iristectorigen (3 OH groups) concentration in the aqueous extract of *Crocus* leaves was slightly higher (0.073 mg/g) than that in the ethanol (0.058 mg/g). Nigricin (a compound that has 1 OH-group and 2 OCH_3_-groups) was only found in the aqueous extract of the leaves (0.069 mg/g). Biochanin A was only identified in the aqueous extract of *Crocus* leaves (0.108 mg/g). Irigenin (3 OH-group, 3 OCH_3_ groups), like the other isoflavonoids, was found in the extracts of *Crocus* leaves for the first time; however, it showed the lowest concentration among all identified substances (0.028 mg/g and 0.031 mg/g for ethanolic and aqueous extracts, respectively).

It was noted that the flavonoid glycoside isoorientin showed a higher concentration (8.51 mg/g) in comparison with other components. Flavonoids are potent antiproliferative and cytotoxic agents [37]. Their composition and concentration significantly contributed to the cytotoxic activity in *Crocus* extracts [38].

The content of all compounds depends on climatic conditions, cultivation methods, drying, and storage [6]. Smolskaite et al. [39] analyzed 46 samples of *Crocus* leaves from Azerbaijan, India, New Zealand, Morocco, France, Turkey, Iran, Italy, and Spain by HPLC–UV and HPLC–MS methods. Several flavonoids were identified including kaempferol-8-C-gluco-6,3-O-diglucoside and kaempferol-8-C-gluco-6-O-glucose, which were detected for the first time. The authors noted that different samples of *Crocus* leaves did not differ in chemical composition but varied in the concentration of each component. The current research focused on the analysis of *Crocus* from Ukraine, as it is a new country in the saffron market. It should be noted that the quality of saffron grown in Ukraine is high and meets the highest ISO category [6]. This indicates an increased interest in this crop, especially since the climatic conditions of Ukraine are well suited for the cultivation of this plant. The increased demand for the cultivation of saffron results in the production of large quantities of its by-products, especially the leaves. Thus, it is important to evaluate the economic value of the leaves in terms of their biologically active secondary metabolites.

### 2.2. HPLC Method Validation

An HPLC method was developed for the simultaneous quantitative determination of phenolic compounds in *Crocus* leaf extracts. Full validation of this method was carried out. Method validation was performed to demonstrate the applicability of the developed analytical method. The following parameters were tested according to the guidelines of the International Conference on Harmonization (ICH) Q2 (R1): specificity, linearity, the limit of detection (LOD), the limit of detection (LOQ), accuracy, and precision. The results are given in Appendix A. The standard curves showed good linearity (R^2^ > 0.999) over a certain concentration range for each reference standard. The relative standard deviation (RSD) of the peak areas of the sixteen reference standards was less than 1.6%, suggesting that the instrument was in good condition. The intra- and inter-day precision and accuracy were less than 1.35% and 1.57%, respectively. The accuracy of the tested compounds was within the range of 98.08–102.88%. The LOQs and LODs were also acceptable for the simultaneous determination of all reference components, indicating that the developed HPLC method was satisfactory for the quantitative analysis of *Crocus* extracts.

### 2.3. Cytotoxic Activity of the Extracts

All tested extracts showed cytotoxic activity against the melanoma (IGR39) and triple-negative breast cancer (MDA-MB-231) cells (Figure 2). The ethanolic extract of *Crocus* leaves was more active than the aqueous extract for both cell lines (EC_50_ of the ethanolic extract against IGR39: 0.41 ± 0.10 mg/mL, against MDA-MB-231: 0.33 ± 0.04 mg/mL; EC_50_ of the aqueous extract against IGR39: 0.57 ± 0.11 mg/mL, against MDA-MB-231: 1.13 ± 0.09 mg/mL). It was found that the aqueous extracts significantly reduced the viability of melanoma (IGR39) cells compared with MDA-MB-231 cells.

The cytotoxic activity of the tested ethanolic extracts against breast cancer cell lines was generally higher than that of the corresponding aqueous extract (Figure 2). The potent cytotoxic activity is generally associated with the presence of various phenolic compounds.

### 2.4. Molecular Docking Studies

The molecular docking study aimed to analyze the biological activity of the identified compounds in *Crocus* extracts. In addition, it aimed to investigate the descriptors with the highest cytotoxic activity and to examine the possible mechanism of action of the selected compounds. Among the identified compounds, kaempferol, rutin, mangiferin, and quercetin were suitable for selected molecular docking experiments according to the literature data [40,41,42].

Several studies indicated that *Crocus* extracts exhibited cytotoxic effects because of the presence of crocin, picrocrocin, and safranal as well as phenolic compounds [15,30,31]. Kaempferol showed an antiproliferative effect on human breast carcinoma MDA-MB-453 cells [43], the mouse melanoma cell line B16F1 [44], and A375 human malignant melanoma cells [45]. To assess the most potent compounds for further studies, molecular docking experiments were performed to evaluate the affinity of individual components of the extracts in silico, as well as to investigate the activity of the most promising compounds on cancer cell lines.

For docking research on cytotoxic activity, scientists use different types of proteins depending on the nature of cancer [46]. For instance, the breast cancer proteins with BRCA1-type HRD phenotypes such as 1T15 [47], 4Y2G [48], 4RJ3 [49], estrogen receptors 2IOK [49,50,51] and 2IOG [52]; aromatase 4GL7 [53], or oncogenic kinases/phosphatases (JAK2 [54]) are the most commonly used targets for molecular docking research. Lyu et al. [55] stated the importance of human epidermal growth factor receptor HER3 in the pathogenesis of metastatic breast cancer. The enzymes chosen for docking studies were selected based on their reported binding energy for natural compounds, especially flavonoids. The interaction with c-Met kinases (PDB codes: 2RFN and 4XYF) [56] is a key mechanism of rutin’s cytotoxic activity against breast cancer. HSP90 proteins (PDB ID 4RJ3, 2IOK) were used for in silico docking analysis of curcumin and resveratrol on breast cancer proteins compared with their effect on the MCF-7 cell line [49]. Among estrogen receptors that play an important role in cancer pathogenesis, we selected the human ERα-LBD (PDB ID 3ERT) complex with 4-hydroxytamoxifen, which is an active metabolite of tamoxifen. The affinity of this enzyme to naturally occurring flavonoids such as quercetin, luteolin, myricetin, kaempferol, naringin, hesperidin, galangin, baicalein, and epicatechin was investigated as one of the factors that suppress MCF-7 cell line growth [54].

According to the docking studies, several substances identified in the active extracts showed affinity to the active sites of the selected enzymes that were similar or even better than their native ligands (Table 3).

According to the docking results, it was found that almost all the natural compounds from different chemical groups showed an affinity for the selected enzymes (expressed as a negative score function). Interestingly, some of the constituents of *C. sativus* extract including chlorogenic acid, isoorientin, ferulic acid, tectoridin, quercetin, cinnamic acid, genistein-7-glucoside, apigenin, kaempferol, and irigenin showed, better affinity compared with the native ligands of breast cancer proteins 4RJ3, 2IOK, and 4XYF. Moreover, the affinity of many compounds (chlorogenic acid, ferulic acid, cinnamic acid, and iristectorigenin B) to the estrogen receptor (3ERT) was comparable to hydroxytamoxifen. However, the role of each compound should be proven at least by in vitro studies. Among the constituents identified in the most active ethanolic leaf extract, we selected certain compounds for the in vitro studies, including phenolic acids—cinnamic and caffeic—as well as flavonoids—tectoridin, kaempferol, nigricin, irigenin, and mangiferin.

### 2.5. Cytotoxic Activity of Secondary Metabolites from C. sativus Leaf Extracts

The ethanolic extract of *Crocus* leaves showed higher cytotoxic activity than the aqueous extract against both cell lines. This could be due to higher levels of mangiferin (2.031 ± 0.648 mg/g), tectoridin (0.107 ± 0.045 mg/g), ferulic acid (0.251 ± 0.005 mg/g), caffeic acid (1.425 ± 0.012 mg/g), quercetin (0.255 ± 0.003 mg/g), nigricin (0.069 ± 0.003 mg/g), and kaempferol (0.046 ± 0.002 mg/g) in the leaves’ ethanolic extract (Table 1) since these compounds showed cytotoxic activity in various cell lines. Grasso R. et al. [57] showed that ferulic acid reduced the viability of U-87 cells by up to 70% in comparison with the control, and the cells treated with ferulic acid in combination with NLCs (nanostructured lipid carriers) showed a reduction in viability up to 40%. In vitro studies showed that flavonoids quercetin and kaempferol selectively suppressed the viability of brain, breast, cervix, and lung cancer cells [58,59]. Yamagata K. et al. demonstrated that chlorogenic acid significantly suppressed the viability of A549 cancer cells and regulated the expression of genes associated with apoptosis [60]. According to previous literature, caffeic acid showed a cytotoxic effect, which was attributed to its antioxidant properties by suppressing the formation of ROS and DNA damage [61]. The cytotoxic activity of the water and ethanolic extracts of *Crocus* leaves is affected by the concentration of the active substances. The higher levels of flavonoids, phenolic acids, and their derivatives in the ethanol extract of the leaves can be correlated with the more potent cytotoxic effect of this extract compared with the aqueous extract.

To evaluate the effect of individual compounds on the cytotoxic activity of the whole extracts, several compounds, such as quercetin, kaempferol, mangiferin, rutin, apigenin, caffeic acid, irigenin, nigericin, and tectoridin, were tested against melanoma and triple-negative breast cancer cell lines (Figure 3).

Kaempferol and quercetin were the most active compounds, but their concentrations in the extracts were not very high. Therefore, it can be assumed that the activity of extracts was due to the synergistic activity of all its cytotoxic compounds.

### 2.6. Antioxidant Activity

Six compounds with antioxidant properties were identified in the *Crocus* leaf extracts, including mangiferin, isoorientin, caffeic acid, ferulic acid, rutin, and tectoridin, by post-column HPLC accompanied with the ABTS test (Table 4). The antioxidant activity values of the leaf aqueous and ethanol extracts were 0.635 and 0.951 mmol/L, respectively, and were comparable to Trolox (0.3995 mmol/L). The leaves’ bioactivity was correlated to the presence of the following two antioxidant phenolic compounds: mangiferin and rutin. The structure of the phenolic compounds, especially the number and position of the hydroxy groups and the nature of the substitutions on the aromatic ring, significantly affects the activity of the compounds [62,63,64]. Our findings indicated that *Crocus* leaves, a by-product of saffron production, can serve as a valuable natural source of antioxidants.

### 2.7. Bioactivity Screening and Anti-Neuraminidase Activity

There is a lack of studies on the anti-neuraminidase, anti-inflammatory, anti-allergic and antiviral activities of the *C. sativus* leaf extracts. According to our results (Table 5), the aqueous extract at 10 µg/mL inhibited superoxide anion generation by 35.67 ± 5.62% and elastase release by 19.62 ± 2.90% in fMLF/CB-induced human neutrophils. Moreover, both the aqueous and ethanolic extracts increased the effect of NRF2 expression exclusively in normal cells (152.5% and 136.5%, respectively), while the NRF2 expression was intact in the cancer cell line at 100 µg/mL (106.2% and 104.5%, respectively). Previous studies indicated that plant extracts rich in phenolic compounds may affect the antioxidant function of cells, including NRF2 expression [65]. The *C. sativus* leaf aqueous and ethanolic extracts were inactive in the other bioactivity assays (Table 5), including the anti-allergic (degranulation assay, 100 µg/mL), anti-viral (influenza H1N1 and enterovirus D68, 50 µg/mL; ethanolic extract on coronavirus 229E, 10 µg/mL), and lipid droplets assay (100 µg/mL). Our results, along with the literature data, suggest that the high content of the phenolic compounds may correlate with the inhibitory effects of *C. sativus* leaves on superoxide anion generation in neutrophils and NRF2 activation in normal cells. A higher concentration may be needed for other bioactivity assays to show positive results.

## 3. Materials and Methods

### 3.1. Chemicals and Reagents

The acetonitrile and methanol were of HPLC grade and were purchased from Roth GmbH (Karlsruhe, Germany). The reference compounds (chlorogenic acid, caffeic acid, ferulic acid, *t*-cinnamic acid, mangiferin, kaempferol, tectoridin, iristectorigenin B, nigricin, irigenin, isoorientin, genistein-7-glucoside, and biochanin A) were purchased from ChromaDex (Santa Ana, CA, USA), Sigma-Aldrich (Saint Louis, MO, USA), HWI ANALYTIK GmbH, and Roth GmbH (Karlsruhe, Germany). All other chemicals were of analytical grade.

### 3.2. Crude Extracts Preparation

*C. sativus* leaves were collected from the plantation in Lyubimivka village (Ukraine) in November 2019. Botanical identification was performed by Dr. Mykhailenko, National University of Pharmacy, Ukraine. The leaves were air-dried at room temperature with suitable ventilation and were crushed to grit (1.0 mm size) for further extraction. The dried material (100 g) was extracted with boiled distilled water (1 L) or ethanol/water 70/30 (*v*/*v*) at 90 °C for 2 h three times on a horizontal water bath shaker. The extracts were filtered through Whatman no. 40 filter paper (Whatman International Ltd., Kent, UK) using a Büchner funnel. The filtered solution was evaporated to dryness under reduced pressure at 40 °C at an approximate rotation speed of 200 rpm using a rotary evaporator (Heidolph, Germany) to obtain a crude extract (18.8 g and 20.5 g for the water and ethanolic extracts, respectively). Finally, the extracts were refrigerated at 4 °C in an airtight container until further analysis. Before the bioactivity assay, the crude extracts were dissolved in DMSO (10% final concentration) and were filter-sterilized with a 0.2 µm membrane filter (Millex) to prepare a stock solution.

### 3.3. Sample Preparation for HPLC Analysis

A portion of the dry plant extract (0.1 g) was extracted with 10 mL methanol in an ultrasonic bath at room temperature (20 ± 2 °C) for 30 min. The solutions were filtered through a membrane filter (0.45 µm) before use. An aliquot of 10 µL was injected into the HPLC system for analysis. A standard solution of the reference compounds at a concentration of 1.0 mg/mL for each compound was prepared by dissolving it in methanol and was used for calibration. All samples were kept at 4 °C before use.

### 3.4. HPLC Conditions

The Shimadzu Nexera X2 LC-30AD HPLC system (Shimadzu, Kyoto, Japan) includes a quaternary pump, an online degasser, a column temperature controller, the SIL-30AC autosampler, and SPD-M20A diode array detector (DAD) (Shimadzu, Kyoto, Japan) was used for fingerprint analysis. The chromatographic separation of the phenolic compounds was carried out using an ACE C_18_ column (250 mm × 4.6 mm, 5.0 μm; Bellefonte, PA, USA). The mobile phase was composed of solvent A (0.1% acetic acid in water) and solvent B (acetonitrile). The elution was performed at a flow rate of 1 mL/min, and the following linear gradient program was applied: 0–8 min, 5–15% B; 8–30 min, 15–20% B; 30–48 min, 20–40% B; 48–58 min, 40–50% B; 58–65 min, 50% B; 65–66 min, 50–95% B. The column temperature was maintained at 25 °C. The injection volume of the sample solution was 10 μL. The detection wavelengths were 270 nm, 240 nm, and 310 nm. The standard solutions of the reference compounds were used for the calibration of a standard curve using an external standard method. The analyses were performed in duplicate. Chromatographic peak identification was carried out according to the analyte and reference compound retention time by comparing the UV absorption spectra of the reference compounds and analytes obtained with a diode array detector.

The validation of the HPLC method was performed according to the guideline of the International Conference on Harmonization (ICH) and included the following parameters: specificity, linearity, precision, LOD, and LOQ (Appendix A).

### 3.5. HPLC–PDA, and UPLC–MS Conditions, HPLC Post-Column Assay

HPLC–PDA and HPLC–ABTS were done using a Waters Alliance 2695 separation module system as previously described by Marksa et al. [66] with some modifications. UPLC–MS analysis was performed using the ACQUITY H-class UPLC system (Waters, Milford, MA, USA). The details are described in the Appendix A.

### 3.6. In Vitro Cytotoxic Activity

The human melanoma cancer cell line IGR39 and the human triple-negative breast cancer cell line MDA-MB-231 were used, and the method is described in detail in the Appendix A. These cell lines were obtained from American Type Culture Collection (ATCC, Manassas, VA, USA).

### 3.7. Antiallergic Activity in RBL-2H3 Cells

The methylthiazole tetrazolium (MTT) assay was used to measure the possible toxic effect of the samples on RBL-2H3 cells as previously described [67]. The *C. sativus* leaf aqueous and ethanolic extracts were then subjected to an anti-allergic degranulation assay based on *β*-hexosaminidase release in RBL-2H3 mast cells induced by calcium ionophore (A23187) or antigen (IgE plus DNP-BSA) according to the previous methodology [68,69]. The details of the assays are presented in the Appendix A.

### 3.8. Anti-Inflammatory Activity in Human Neutrophils

Blood was taken from healthy human donors using a protocol approved by the Chang Gung Memorial Hospital review board. Neutrophils were isolated according to the standard procedure described previously [70]. The inhibition of superoxide anion generation was measured by the reduction of ferricytochrome *c*, as previously described [71]. Elastase release representing degranulation from azurophilic granules was evaluated as described previously [72]. Details can be found in the Appendix A.

### 3.9. Lipid Droplet Assay

The lipid droplet assay was performed according to a previous method using a BSA-conjugated oleic acid system in Huh7 cells [73]. The details are described in the Appendix A.

### 3.10. NRF2 Activity

Nuclear transcription factor NRF2 activity was evaluated in HaCaT normal cells and Huh7 cancer cells according to previous methodology [74]. The cell line HaCaT/ARE carrying the antioxidant response element was developed using an HaCaT stable cell line carrying a fragment derived from the pGL4.37 (luc2P/ARE/Hygro) plasmid and the luciferase reporter gene luc2P. The details are described in the Appendix A.

### 3.11. Protective Effect of the Extracts against Influenza Virus and Enterovirus

The anti-viral assay was performed by cytopathic effects of the extracts on the cells infected by influenza H1N1 [75], and enterovirus D68 [76]. The details are described in the Appendix A.

### 3.12. Coronavirus 229E Assay

The protective effect of the samples against human coronavirus 229E (HCoV-229) was determined according to a previously described method [76]. The details are described in the Appendix A.

### 3.13. Neuraminidase Activity Assay

A baculovirus displayed neuraminidase NA9 on the surface (NA9-Bac) as a pseudotyped influenza virus was used to evaluate the neuraminidase activity. Details are described in [77] and in the Appendix A.

### 3.14. Molecular Docking

The docking simulations were performed with the SCIGRESS software package (Fujitsu, Fukuoka, Japan (license 742F6852C191)).

#### 3.14.1. Ligand Preparation

All structures were designed using the ISIS DRAW 4.0 software and were saved as .mol files. The structures were imported into the SCIGRESS software and were saved in .csf format. Hydrogen atoms were added. Structure optimization was performed using the MM3 method.

#### 3.14.2. Enzymes Preparation

The X-ray crystal structures of proteins were obtained from the Protein Data Bank (http://www.rcsb.org/pdb, accessed 15 September 2021). Hydrogen atoms were added, and structure optimization was carried out. Water and heteroatoms were removed from the structure. The native ligand was removed from the active site after its identification.

Genetic algorithm (GA) was used within automatic docking (Population size: 500; Crossover: 0.2; Elitism: 50; Max generations: 50.000; Mutation rate: 0.0; Convergence 0.1; Max iterations: 1000; and Rate: 0.01). Docking studies were carried out using fast quantum molecular docking (rigid receptor and flexible ligand). The calculated score functions of the substances investigated were compared with the ones obtained for the native ligand. Water molecules were removed, and hydrogens were added to the crystal structure of proteins before docking. Assigning charge and protonation state final refinement (energy minimization) were conducted using MM3 force field runs.

### 3.15. Data Analysis

The processing of HPLC data was carried out using the LabSolutions Analysis Data System (Shimadzu, Kyoto, Japan). The results were analyzed by one-way analysis of variance (ANOVA) followed by Tukey’s multiple comparison test with the software package Prism v. 5.04 (GraphPad Software Inc., La Jolla, CA, USA, chemical composition means ± SD), by Dunnet’s test (GraphPad Prism 6.0, GraphPad Software Inc., San Diego, CA, USA, anti-allergic assay; data of cytotoxicity and antioxidant assays are presented as means ± standard error (S.E.) of at least three independent experiments), or Student’s *t*-test (SigmaPlot, Systat Software Inc., San Jose, CA, USA, anti-inflammatory assay, means ± S.E.M.). Values with *p*-values below 0.05 were considered statistically significant. At least three independent measurements were performed unless otherwise specified.

## 4. Conclusions

The current study investigated the in vitro cytotoxic and antioxidant activity of *C. sativus* leaf extracts. A quality control method using a validated HPLC fingerprint analysis for *C. sativus* leaf extracts was developed. Further, the molecular docking studies indicated a good binding between the identified compounds and breast cancer- and melanoma-related enzymes. The cytotoxic and antioxidant activities of the major identified compounds correlated well with the bioactivities of the whole extracts. Our results indicated that the leaves of *C. sativus* are a rich source of biologically active compounds that can have potential therapeutic applications and should not be wasted during the saffron preparation process.

## Figures and Tables

**Figure 1 molecules-27-00010-f001:**
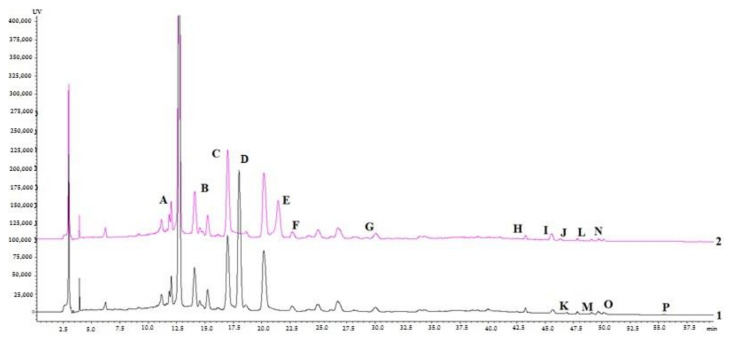
HPLC–DAD chromatograms of *C. sativus* leaves aqueous (black line 1) and ethanolic (70%, *v*/*v*) (pink line 2) dry extracts: Chlorogenic acid (A); Caffeic acid (B); Mangiferin (C); Isoorientin (D); Ferulic acid (E); Rutin (F), Tectoridin (G); Quercetin (H); Cinnamic acid (I); Nigricin (J); Genistein-7-glucoside (K); Apigenin (L); Kaempferol (M); Iristectorigenin B (N); Irigenin (O); Biochanin A (P).

**Figure 2 molecules-27-00010-f002:**
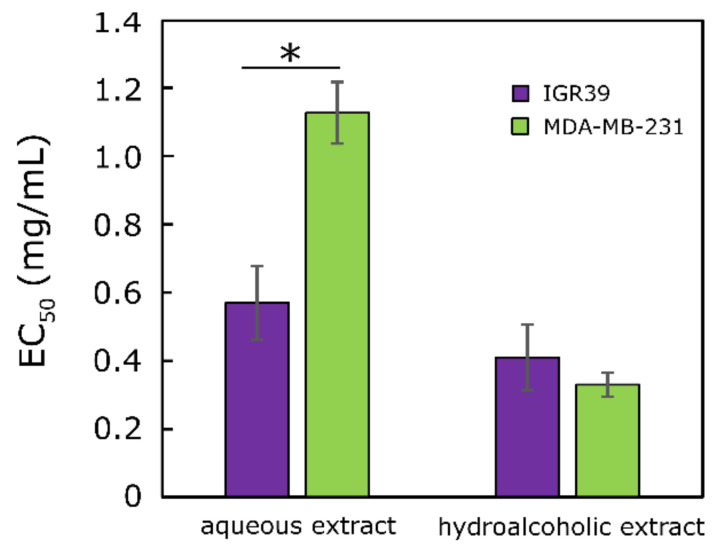
EC_50_ values of *C. sativus* leaf extracts against MDA-MB-231 and IGR39 cell lines after 72 h. * *p* < 0.05, *n* = 3.

**Figure 3 molecules-27-00010-f003:**
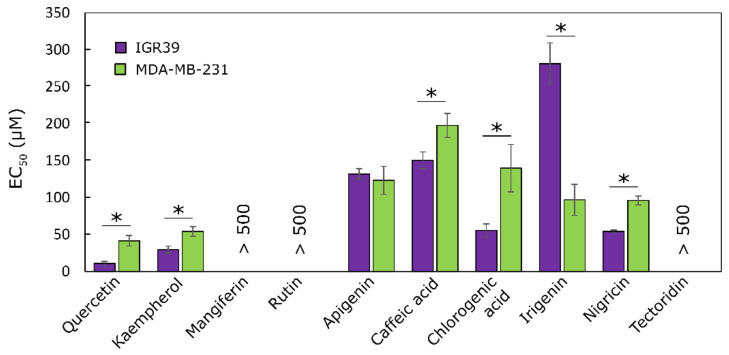
Cytotoxic activity of the tested individual compounds against IGR39 and MDA-MB-231 cell lines, * *p* < 0.05, *n* = 3.

**Table 1 molecules-27-00010-t001:** Secondary metabolite contents (mg/g dry weight) of the aqueous and ethanolic dry extracts of *C. sativus* leaves.

№	Compounds	Rt, min/*λ*, nm	Content, mg/g	
			Aqueous Extract	Ethanolic Extract
A	Chlorogenic acid	11.66/310	0.677 ± 0.037	0.678 ± 0.004
B	Caffeic acid	14.18/310	-	1.425 ± 0.012
C	Mangiferin	17.18/270	1.823 ± 0.124	2.031 ± 0.648
D	Isoorientin	17.66/310	8.508 ± 0.001	-
E	Ferulic acid	21.64/310	-	0.251 ± 0.005
F	Rutin	22.48/310	0.085 ± 0.027	0.100 ± 0.005
G	Tectoridin	29.69/270	0.044 ± 0.032	0.107 ± 0.045
H	Quercetin	43.71/310	-	0.255 ± 0.003
I	*t*-Cinnamic acid	45.22/270	0.106 ± 0.001	0.333 ± 0.006
J	Nigricin	45.50/270	-	0.069 ± 0.003
K	Genistein-7-glucoside	46.07/270	0.407 ± 0.015	-
L	Apigenin	47.90/340	-	0.088 ± 0.005
M	Kaempferol	48.96/310	0.033 ± 0.003	0.046 ± 0.002
N	Iristectorigenin B	49.15/270	0.073 ± 0.025	0.058 ± 0.003
O	Irigenin	50.03/270	0.031 ± 0.016	0.028 ± 0.001
P	Biochanin A	55.85/270	0.108 ± 0.325	-

**Table 2 molecules-27-00010-t002:** Chromatographic, UV, and mass spectroscopic data of the reference compounds.

Retention Time, min (UPLC–MS)	Compound	UV *λ*_max_ (nm)	Mol. Formula	Mol. Weight, g/mol	[M−H]−(*m*/*z*)	Fragment Ions (−)
3.69	Chlorogenic acid	218, 241, 327	C_16_H_8_O_9_	354.31	353	191, 179, 135
3.92	Caffeic acid	217, 236, 324	C_9_H_8_O_4_	180.16	179	161, 135
4.21	Mangiferin	240, 318, 257, 365	C_19_H_18_O_11_	422.30	421	403, 331, 301, 259, 271
4.51	Isoorientin	269, 349	C_21_H_20_O_11_	448.38	447	429, 411, 327, 297, 285
4.64	Genistein-7-glucoside	259, 332	C_21_H_20_O_10_	432.37	431	239, 268, 269, 311, 431
4.82	Rutin	255, 352	C_27_H_30_O_16_	610.52	609	301
5.09	Ferulic acid	218, 235, 323	C_10_H_10_O_4_	194.18	193	178, 149,134
6.22	Apigenin	237, 267, 337	C_15_H_10_O_5_	270.24	269	158
6.54	Tectoridin	263, 328	C_22_H_22_O_11_	462.41	461	446, 411, 341, 298
6.74	Quercetin	254, 369	C_15_H_10_O_7_	302.24	301	273, 227, 179, 151, 93
6.80	*trans*-Cinnamic acid	322, 276	C_9_H_8_O_2_	148.16	147	119, 103
7.41	Kaempferol	265, 365	C_15_H_10_O_6_	286.24	285	239, 187
7.46	Iristectorigenin B	218, 265	C_17_H_14_O_7_	330.29	329	314, 311, 299, 271, 255
7.58	Irigenin	264, 218	C_18_H_16_O_8_	360.31	359	344, 329, 314, 286, 258
8.20	Biochanin A	262, 345	C_16_H_12_O_5_	284.26	283	268, 267, 239, 211, 132
ND *	Nigricin	262, 322	C_17_H_12_O_6_	312.28	ND	ND

* ND: compound was not detected in negative ion mode.

**Table 3 molecules-27-00010-t003:** Binding energies of the selected compounds identified from *C. sativus* leaf extracts docked against the target protein receptors.

		Binding Energies in the Active Site, kcal/mol			
№	Compounds	Breast Cancer Proteins			Melanoma
		4RJ3	2IOK	4XYF	3ERT
1	Chlorogenic acid	−70.448	−70.910	−95.773	−84.727
2	Caffeic acid	−103.721	−73.071	−73.432	−79.224
3	Mangiferin	−36.321	−74.476	−93.985	−72.074
4	Isoorientin	−72.180	−50.336	−103.029	−68.522
5	Ferulic acid	−74.705	−90.033	−75.126	−90.066
6	Rutin	−59.391	−62.756	−53.408	−75.257
7	Tectoridin	−88.706	−78.872	−96.284	−70.155
8	Quercetin	−77.893	−56.916	−94.328	−66.201
9	*t*-Cinnamic acid	−81.085	−72.741	−60.426	−84.200
10	Genistein-7-Glu	−76.241	−67.142	−96.593	−72.092
11	Apigenin	−87.532	−73.515	−82.298	−68.500
12	Kaempferol	−90.462	−71.603	−86.092	−79.633
13	Iristectorigenin B	−79.516	−75.179	−55.603	−86.510
14	Nigricin	−83.299	−84.250	−74.943	−81.789
15	Irigenin	−86.146	−80.691	−80.050	−56.689
	**Native ligands**				
	Ligand 4RJ3	−86.564			
	Ligand 2IOK		−69.486		
	Hydroxytamoxifen				−83.083
	Ligand 4XYF			−75.090	

**Table 4 molecules-27-00010-t004:** The radical scavenging activity of individual compounds from *C. sativus* leaf extracts expressed as TEAC (mmol/L) using the ABTS post-column assay.

Component	Retention Time	Aqueous Extract	Ethanolic Extract
Caffeic acid	14.283	–	0.172 ± 0.008
Mangiferin	15.313	0.405 ± 0.017	0.384 ± 0.017
Isoorientin	18.693	0.003 ± 0.001	–
Ferulic acid	21.870	–	0.162 ± 0.007
Rutin	22.398	0.224 ± 0.010	0.209 ± 0.009
Tectoridin	29.759	0.003 ± 0.0001	0.024 ± 0.001
Total		0.635 ± 0.007	0.951 ± 0.010

**Table 5 molecules-27-00010-t005:** Neuraminidase (NA) and lipid droplets activity assays.

Sample	Relative NRF2 Activity ^a^ in HacaT Cells ^b^(%, mean ± SD)	Relative NRF2 Activity ^a^ in Huh7 Cells ^b^(%, mean ± SD)	NA9 Inhibition Activity ^c^(%, mean ± SD)	Lipid Droplet Inhibition Activity ^d^(%, mean ± SD)	Superoxide Anion Generation, Human Neutrophils ^e^ (%, mean ± SEM)	Elastase Release, Human Neutrophils ^e^ (%, mean ± SEM)	A23187-Induced Degranulation Assay, RBL-2H3 Cells ^f^ (%, mean ± SD)	Antigen-Induced Degranulation Assay, RBL-2H3 Cells ^f^ (%, mean ± SD)	Protective Activity against Influenza H1N1, MDCK Cells ^g^	Protective Activity against Enterovirus 68, RD Cells ^g^	Protective Activity against Coronavirus 229E, Huh7 Cells ^g^
*C. sativus* leaf aqueous extract	152.5	106.2	7.4 ± 2.9	86.3 ± 17.6	35.67 ± 5.62 **	19.62 ± 2.90 **	8.3 ± 7.4	2.0 ± 3.5	inactive	inactive	–
*C. sativus* leaf ethanolic extract	136.5	104.5	6.8 ± 1.9	101.3 ± 17.3	– ^l^	– ^l^	6.3 ± 5.1	13.7 ± 1.5	inactive	inactive	inactive
TBHQ ^h^	684.3 ± 53.3	–	–	–	–	–	–	–	–	–	–
Luteolin ^i^	–	23.8 ± 0.3	–	–	–	–	–	–	–	–	–
Zanamivir ^j^	–	–	97.4 ± 0.0	–	–	–	–	–	–	–	–
TC ^k^	–	–	–	16.3 ± 0.2	–	–	–	–	–	–	–

^a^ Relative luciferase activity (NRF2) was calculated by normalizing luciferase activity to cell viability and presented as the fold to solvent control (*n* = 1). *C. sativus* leaves 100 µg/mL. ^b^ HacaT, a normal skin cell line. Huh7, a liver cancer cell line. ^c^ Neuraminidase inhibition assay (*n* = 1). *C. sativus* leaves 100 µg/mL. ^d^ Lipid droplet count, the average LD counts/cell of OA were used as a standard for 100% of fatty loading in the Huh7 cell line (*n* = 1). Saffron leaves 100 µg/mL. ^e^ Results are presented as means ± SEM (*n* = 3) compared with the control (fMLF/CB), ** *p* < 0.01. Genistein served as the positive control and inhibited 99.7% of superoxide anion generation at 10 μg/mL and 101.2% of elastase release at 30 μg/mL [30]. *C. sativus* leaves 10 µg/mL. ^f^ The cytotoxicity of the sample was evaluated by MTT assay (95.0 ± 8.7%). Inhibition of *β*-hexosaminidase release was evaluated, and the results are presented as means ± SD (*n* = 3) compared to the untreated control (DMSO). Dexamethasone (10 nM) was used as a positive control and inhibited 65.7 and 66.3% of A23187- and antigen-induced *β*-hexosaminidase release, respectively [30]. *C. sativus* leaves 100 µg/mL. ^g^ The protective effects were evaluated based on the viability of cells infected by the virus. Saffron stigmas 50 µg/mL for influenza and enterovirus (*n* = 2), 10 µg/mL for coronavirus 229E assay (*n* = 1). Inactive, no significant inhibition. ^h^ TBHQ, 2-(1,1-dimethylethyl)-1,4-benzenediol was used as positive control for Nrf2 activation. The drug concentration was 10 µM. ^i^ Luteolin was used as a negative control for Nrf2 activation. The drug concentration was 50 µM. ^j^ Zanamivir was used as a positive control for NA inhibition. The drug concentration was 1 µM. ^k^ TC, Triacsin C, is an inhibitor of long fatty acyl CoA synthetase and was used as a positive control for lipid droplet inhibition. The drug concentration was 1 µM. ^l^ The sample was not soluble. –, not tested.

## Data Availability

This research contains only Appendix A.

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
