# Peer review of "Pharmacological Potential and Chemical Composition of Crocus sativus Leaf Extracts"

_molecules, 2021, doi:10.3390/molecules27010010_

Round 1

Reviewer 1 Report

The manuscript describes the chemical composition of the aqueous and ethanolic extracts of C. sativus leaves and their cytotoxic and antiviral activity against several cell line assays. It’s important to evaluate the pharmacological potential of the bioactive compounds contained in the C. sativus by-products such as leaves which represents an excellent source of bioactive compounds.

The recommendation is "Reconsider after major revision". The authors should revise the manuscript in the following points, and resubmit the revised manuscript:

  1. The characterization of the major components of the Crocus leaves was only done based on the chromatographic parameters, i.e., Rt. It would be necessary that the authors also carry out a second-level identification based on the mass spectrometric analysis, thus providing the molecular weight (MW) information of the components. Identification based on the measurement of the MW, preferably the exact MW, is vital in the process of the following evaluation of their cytotoxic and antiviral activity. Therefore, the authors should include a Table with the HRMS-derived MW information of the major components of the Crocus leaves.
  2. The authors have previously published the characterization of phytochemical components of Crocus sativus leaves (Sci. Pharm. 2021, 89, 28), and a similar chromatographic separation was shown in that publication. Nonetheless, the provided mass spectrometric information was mainly MS/MS fragmentation data. In the current study, it would be advisable that the authors provide the accurate MW information of the Crocus components. That would also differentiate this study from the previous one.

Several points in the manuscript should be revised and corrected as follows:

  1. Line 47: “Potent pharmacological potential” should read: “Potent pharmacological activity”.
  2. Line 101: “The use of these byproducts” should read: “The exploitation of these byproducts”.
  3. In Figure 1, the annotation D should be moved over the peak at Rt66.
  4. Line 172: mangiferin is peak C (and not 3) and it is eluted at Rt18 NOT 14.18.
  5. Line 227: It should be R2 > 0.999
  6. Line 237: Data for U87 cell lines are not shown in Figure 2. The authors should provide a comment if the extracts do not have cytotoxic activity for this cell line.
  7. In Table 2 correct Kempferol with Kaempferol.
  8. Lines 289-291: According to Table 2, Genistein-7-Glu has higher affinity (-96.59 kcal/mol) to 4XYF protein than the Ligand 4XYF (-75.090 kcal/mol), whereas Cinnamic acid has higher affinity (-72.74) kcal/mol) to 2IOK protein than the Ligand 2IOK (-69.48 kcal/mol) and higher affinity (-84.2 kcal/mol) to 3ERT than hydroxytamoxifen (-83.08 kcal/mol). Similarly, Quercetin has higher affinity (-94.32kcal/mol) to 4XYF protein than the Ligand 4XYF (-75.090 kcal/mol). The authors should rephrase and/or provide explanation on these discrepancies.
  9. Line 296: This sentence should be removed as it is stated in the previous sentence.
  10. The authors should replace “water extract” with “aqueous extract” in several parts of the text.
  11. Lines 350 and 358: Table 3 should be corrected to Table 4. Similarly, Table 3 should be corrected to Table 4 in Line 724.
  12. Lines 513: The authors have to rephrase the last sentences in the Conclusions part and try not to be repetitive.

The authors should address and correct the aforementioned points and resubmit the revised manuscript.

Author Response

The manuscript describes the chemical composition of the aqueous and ethanolic extracts of C. sativus leaves and their cytotoxic and antiviral activity against several cell line assays. It’s important to evaluate the pharmacological potential of the bioactive compounds contained in the C. sativus by-products such as leaves which represents an excellent source of bioactive compounds. The recommendation is "Reconsider after major revision". The authors should revise the manuscript in the following points, and resubmit the revised manuscript:

The characterization of the major components of the Crocus leaves was only done based on the chromatographic parameters, i.e., Rt. It would be necessary that the authors also carry out a second-level identification based on the mass spectrometric analysis, thus providing the molecular weight (MW) information of the components. Identification based on the measurement of the MW, preferably the exact MW, is vital in the process of the following evaluation of their cytotoxic and antiviral activity. Therefore, the authors should include a Table with the HRMS-derived MW information of the major components of the Crocus leaves.

Answer:

Identification of all constituents was performed by HPLC analysis by comparing the retention time (Rt), UV spectra of the peaks in samples with those of reference compounds (16 compounds). The purity of peaks was checked by a diode array detector coupled to the HPLC system, comparing UV spectra of each peak with those of reference compounds and/or by examination of the UV spectra.

According to the reviewer recommendation, to confirm the identified compounds, further UPLC-MS/MS analysis of these components was performed in negative ion mode, and their retention time (tR), calculated molecular weights and MS/MS data are shown in Table 2, respectively. The method of UPLC-MS analysis was added to the supplementary materials.

We used ultra-performance liquid chromatography coupled with a mass spectrometry (UPLC-MS) to rapidly and accurately determine 16 phenolic compounds. Considering the fact that the fragmentation of molecules was previously described and the mass analysis was not the main goal of the current study, a brief description and characteristics of our substances, together with references to literature, are presented in the Supplementary material. However, the main text of the article contains a table with the main spectral characteristics of all compounds as shown below.

Table 2. Chromatographic, UV, and mass spectroscopic data of the reference compounds

The authors have previously published the characterization of phytochemical components of Crocus sativus leaves (Sci. Pharm. 2021, 89, 28), and a similar chromatographic separation was shown in that publication. Nonetheless, the provided mass spectrometric information was mainly MS/MS fragmentation data. In the current study, it would be advisable that the authors provide the accurate MW information of the Crocus components. That would also differentiate this study from the previous one.

Answer: Thank you for your recommendations. To our best knowledge and given a limited time for revisions, we have provided complete information on the identification of substances in the extracts.

Several points in the manuscript should be revised and corrected as follows:

Line 47: “Potent pharmacological potential” should read: “Potent pharmacological activity”.

Thank you for your attention. We agree with the reviewer and applied the correction.

Line 101: “The use of these byproducts” should read: “The exploitation of these byproducts”.

Thank you. We corrected the statement.

In Figure 1, the annotation D should be moved over the peak at Rt66.

Thank you for your attention. We made a change in the chromatogram and corrected the assignment "D" to line 1 (aqueous extract) at Rt 17.66, since orientin was found in this particular extract.

Line 172: mangiferin is peak C (and not 3) and it is eluted at Rt18 NOT 14.18.

Thank you for your attention. We made corrections. We sincerely apologize for the errors.

Line 227: It should be R2 > 0.999

Thank you. Correction was made.

Line 237: Data for U87 cell lines are not shown in Figure 2. The authors should provide a comment if the extracts do not have cytotoxic activity for this cell line.

We would like to thank the reviewer for this valuable observation. We accidentally left some information related to the activity determination of extracts in U87 cell line.

However, as we do not have full data for all compounds in this cell line, we were unable to provide the data. Thus we removed all the accidentally left information from our manuscript.

In Table 2 correct Kempferol with Kaempferol.

Thank you. Correction was made.

Lines 289-291: According to Table 2, Genistein-7-Glu has higher affinity (-96.59 kcal/mol) to 4XYF protein than the Ligand 4XYF (-75.090 kcal/mol), whereas Cinnamic acid has higher affinity (-72.74) kcal/mol) to 2IOK protein than the Ligand 2IOK (-69.48 kcal/mol) and higher affinity (-84.2 kcal/mol) to 3ERT than hydroxytamoxifen (-83.08 kcal/mol). Similarly, Quercetin has higher affinity (-94.32kcal/mol) to 4XYF protein than the Ligand 4XYF (-75.090 kcal/mol). The authors should rephrase and/or provide explanation on these discrepancies.

Thank you so much. We sincerely apologize for incorrect interpretation of result. The paragraph was rephrased as follows:

“According to the docking results, it was found that almost all natural compounds from different chemical groups showed affinity to the selected enzymes (expressed as negative score function). Interestingly, some of the C. sativus extract constituents including chlorogenic acid, isoorientin, ferulic acid, tectoridin, quercetin, cinnamic acid, genistein-7-glucoside, apigenin, kaempferol and irigenin showed better affinity compared to the native ligands of breast cancer proteins 4RJ3, 2IOK, or 4XYF. Moreover, the affinity of many compounds (chlorogenic acid, ferulic acid, cinnamic acid, iristectorigenin B) to the estrogen receptor (3ERT) was comparable to hydroxytamoxifen. However, the role of each compound should be proved by at least in vitro studies.”

Line 296: This sentence should be removed as it is stated in the previous sentence.

Thank you. We removed the repetitive sentence.

The authors should replace “water extract” with “aqueous extract” in several parts of the text.

Thank you. Correction was made.

Lines 350 and 358: Table 3 should be corrected to Table 4. Similarly, Table 3 should be corrected to Table 4 in Line 724.

Thank you. Correction was made.

Lines 513: The authors have to rephrase the last sentences in the Conclusions part and try not to be repetitive.

We made corrections and completely rephrased the Conclusion (p.13) as follows:

“Current study investigated in vitro cytotoxic and antioxidant activity of C. sativus leaves extracts. A quality control method using a validated HPLC fingerprint analysis for C. sativus leaves extracts was developed. Further, the molecular docking studies indicated a good binding between the identified compounds and breast cancer- and melanoma-related enzymes. The cytotoxic and antioxidant activities of the major identified compounds correlated well with the bioactivities of whole extracts. Our results indicated that the leaves of C. sativus are rich in biologically active compounds that can have potential therapeutic applications and should not be wasted during the saffron preparation process.”

The authors should address and correct the aforementioned points and resubmit the revised manuscript.

We are grateful to the reviewer for working with our manuscript, appreciating our work, for all the changes and recommendations. We have taken into account all the comments.

Additionally:

For the compound 5,7-Dihydroxy-4ʹ-methoxyisoflavone, throughout the text, the name was changed to Biochanin A, since the trivial names were used for all compounds.

Reviewer 2 Report

Environmental stress, like temperature, is the modulation of secondary metabolism in plants. Why did you collect the leaves in November (cold weather)?
I suggest the same experiments in different seasons.

Author Response

Environmental stress, like temperature, is the modulation of secondary metabolism in plants. Why did you collect the leaves in November (cold weather)? I suggest the same experiments in different seasons.

Answer:

We want to thank the reviewer for working with our manuscript. It is very pleasant that you paid attention to the period of raw materials harvested for research. And you are really right: metabolic processes slow down in autumn and theoretically for Crocus there is a redistribution of substances from leaves and flowers to underground organs (corms).

Saffron is a crop with a very short growing season (Lopez-Corcoles H., Brasa-Ramos A., Montero-García F., Romero-Valverde M., Montero-Riquelme F. Phenological growth stages of saffron plant (Crocus sativus L.) according to the BBCH Scale. Spanish Journal of Agricultural Research, 2015, 13(3):e09SC01, 7 Ñ€.). The period of activity begins when the corms are planted (late August - early September), then the flowering period (second half of October) and leaves appear en masse (early November). The leaves continue to grow until mid-December. Flowering in this period is coming to an end. The plant goes into dormancy by spring. Crocus sativus is an autumn flowering crocus and at other times you cannot prepare leaves or flowers for research. In addition, the climate of Ukraine is continental and snow falls since December. This creates additional difficulties in the procurement of raw materials.

Additionally:

For the compound 5,7-Dihydroxy-4ʹ-methoxyisoflavone, throughout the text, the name was changed to Biochanin A, since the trivial names were used for all compounds.

Round 2

Reviewer 1 Report

The authors have revised the manuscript following the Editor and Reviewers comments.

I am glad to report that the newly revised manuscript (Ms. No. Molecules – 1484075 v2) can be accepted for publication in Molecules.

Minor Corrections:

  1. Line 152: “were exposed” should read: “were determined”.
  2. Line 157: tR should read Rt.

Reviewer 2 Report

Dear Authors,

I appreciate the responses.

Best regards.

This manuscript is a resubmission of an earlier submission. The following is a list of the peer review reports and author responses from that submission.